# Green Aromatic Epoxidation with an Iron Porphyrin Catalyst for One-Pot Functionalization of Renewable Xylene, Quinoline, and Acridine

**DOI:** 10.3390/molecules28093940

**Published:** 2023-05-07

**Authors:** Gabriela A. Corrêa, Susana L. H. Rebelo, Baltazar de Castro

**Affiliations:** LAQV/REQUIMTE, Departamento de Química e Bioquímica, Faculdade de Ciências, Universidade do Porto, Rua do Campo Alegre s/n, 4169-007 Porto, Portugal

**Keywords:** green chemistry, C-H functionalization, epoxidation, iron porphyrin, oxidation catalysis, renewable aromatics

## Abstract

Sustainable functionalization of renewable aromatics is a key step to supply our present needs for specialty chemicals and pursuing the transition to a circular, fossil-free economy. In the present work, three typically stable aromatic compounds, representative of products abundantly obtainable from biomass or recycling processes, were functionalized in one-pot oxidation reactions at room temperature, using H_2_O_2_ as a green oxidant and ethanol as a green solvent in the presence of a highly electron withdrawing iron porphyrin catalyst. The results show unusual initial epoxidation of the aromatic ring by the green catalytic system. The epoxides were isolated or evolved through rearrangement, ring opening by nucleophiles, and oxidation. Acridine was oxidized to mono- and di-oxides in the peripheral ring: 1:2-epoxy-1,2-dihydroacridine and *anti*-1:2,3:4-diepoxy-1,2,3,4-tetrahydroacridine, with TON of 285. *o*-Xylene was oxidized to 4-hydroxy-3,4-dimethylcyclohexa-2,5-dienone, an attractive building block for synthesis, and 3,4-dimethylphenol as an intermediate, with TON of 237. Quinoline was directly functionalized to 4-quinolone or 3-substituted-4-quinolones (3-ethoxy-4-quinolone or 3-hydroxy-4-quinolone) and corresponding hydroxy-tautomers, with TON of 61.

## 1. Introduction

Biomimetic systems allow to reproduce enzyme activity while avoiding expensive enzyme extractions from natural sources and the expensive procedures of bio- and enzymatic catalysis. Iron and manganese porphyrins have shown the ability to mimic the remarkable activity of oxygenase enzymes, such as cytochrome P450, leading to efficient catalytic systems for sustainable oxidation of aromatic substrates, in mild conditions, and with novel reactivity patterns [1,2]. The porphyrin structure, the microenvironment, such as the solvent, co-catalyst, or catalyst support, have shown to play a key role on biomimetic efficiency [3,4].

A remarkable reaction of P450 during metabolism is the epoxidation of polycyclic aromatic compounds (PACs) in peripheral positions [5]. This reaction is not common in chemical systems, where PACs are mostly oxidized on the *meso*-rings to afford phenols, quinones, and analogues [6]. These are also observed using catalytic systems based on polyoxometalates [7], metallophthalocyanines [8], and metalloporphyrins [9].

In recent years, biomimetic aromatic epoxidations have been disclosed using Mn (2,6-dichlorophenyl)porphyrins (MnP) as catalysts (Figure 1A–C), using non-green conditions, and epoxides of naphthalene, anthracene [10], tetracene [11], and acridine [12] have been obtained (Figure 1A).

In some cases, epoxide formation was considered an intermediate step and products resulting from rearrangement of the epoxide ring were obtained, e.g., the *o*-diketone obtained in phenanthrene oxidation (Figure 1B) [10,13]. It should be noted that alkylbenzenes oxidation in the presence of MnP catalytic systems afforded selectively alkyl group oxidation, e.g., toluene was oxidized mainly to benzoic acid and ethylbenzene to acetophenone (Figure 1C) [14].

Interest in producing valued aromatic compounds from renewable sources has grown enormously in recent years, aiming to implement a circular economy and decrease dependence on fossil-based materials as feedstock in the fine chemicals industry [15,16,17].

Promising techniques for the production of platform green aromatics are the catalytic pyrolysis of biomass or wastes [18,19] and gas-phase Diels–Alder condensation of furan derivatives [20], among others. Optimization studies have been directed towards the increased production of the BTX (benzene, toluene, and xylenes) fraction, where xylenes are of major interest, for use as solvents and as intermediates for synthesis [21].

The chemical oxidation of *o*-xylene is described in the literature using harsh conditions, resulting in methyl groups functionalization [22], or degradation/removal from the environment [23]. Still, in biological systems, *o*-xylene is selectively oxidized in the aromatic ring by a diiron monooxygenase [24].

Quinoline and quinolone scaffolds are present in a vast number of natural compounds and pharmacologically active substances, comprising a significant segment of the pharmaceutical market [25]. Much has been achieved in developing greener syntheses of quinoline and its derivatives. In effect, quinoline can be obtained from biomass derivatives, such as glycerol or levulinic acid by reactions with aniline [26,27].

The direct oxidation of quinoline has been studied to obtain its degradation [28] or site-selective oxidation using enzymatic catalysis [29].

Acridine derivatives are also an important class of bioactive compounds with antibacterial and antimalarial activity and these have been studied as therapeutic agents for cancer and Alzheimer’s disease [12]. Acridine can be obtained from non-fossil sources by catalytic pyrolysis of amino acids [30]. Previous studies on biomimetic oxidation of acridine with Mn porphyrins led to direct and unprecedent epoxidation of the peripheral aromatic rings, disclosing the possibility of new functionalization routes (Figure 1A) [12]. However, it would be desirable to obtain greener conditions, namely the substitution of acetonitrile as the solvent and improve product selectivity.

A green metalloporphyrin system for catalytic oxidation was described, using hydrogen peroxide as a green oxidant, producing water as the only byproduct, a highly electron withdrawing iron porphyrin [Fe(TPFPP)Cl] (FePF; Figure 2), and ethanol as a green solvent, without any other additives or co-catalyst [FePF@H_2_O_2__EtOH]. Moreover, improved methodologies for metalloporphyrin synthesis in eco-sustainable conditions have been reported [31]. This system has been effective in the epoxidation of alkenes and aromatic ring hydroxylation, but direct epoxidation of the aromatic ring has not been observed (Figure 1D) [32,33].

The different catalytic activity of the MnPs and FePF has been ascribed to the formation of different active species in the catalytic cycle [3,14]. With the FePF, a hydroperoxyl species [PFe(III)-OOH] has been ascribed as the active oxidant, while an oxo-species is considered the active oxidant in the catalytic cycle of Mn porphyrins [PMn(V)=O] [32].

The present work describes the application of the [FePF@H_2_O_2__EtOH] catalytic system, at room temperature (RT), in the oxidative valorization of the renewable aromatic compounds (Figure 2).

## 2. Results and Discussion

The oxidation of *o*-xylene (**1**), quinoline (**2**), and acridine (**3**) was carried out by progressive addition of H_2_O_2_ at a rate of 0.6 mmol·h^−1^ in ethanol and at room temperature (RT), using the fluorinated iron porphyrin [Fe(TPFPP)Cl] (FePF) as catalyst. Control reactions performed in the absence of catalyst showed no substrate conversion during the catalytic reaction time.

### 2.1. o-Xylene (**1**)

The catalytic oxidation of *o*-xylene afforded two products, the 3,4-dimethylphenol (**1a**) and 4-hydroxy-3,4-dimethylcyclohexa-2,5-dienone (**1b**). The substrate conversion and product selectivity were monitored by GC-FID and the results are summarized in Table 1.

Using catalyst loadings of 0.3 and 0.6 mol %, the xylene conversion was 30% and 80%, respectively. These values are relevant in the context of C-H bond functionalization, where catalyst loadings between 2.5 and 15 mol % are commonly used [34]. The selectivity for the main product **1b** is 86% and is independent of catalyst loading. The kinetic plot of the reaction described in entry 2 (Figure 1) shows a nearly constant yield of **1a** during the reaction time, which indicates it as an intermediate of the final product **1b**. The maximum conversion was reached after 2.5 h of reaction with a turnover number (TON) of 237.

**Table 1 molecules-28-03940-t001:** Green oxidation of *o*-xylene by Fe porphyrin catalysis in ethanol at room temperature ^a^.

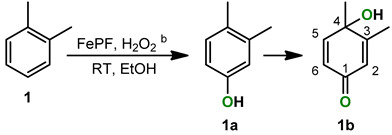						
Entry	[FeP] (mol%)	Time (min)	Conversion (%) ^c^	Selectivity (%) ^c^		TON ^d^
		H_2_O_2_ (eq.) ^b^		1a	1b	
**1**	0.3	90 3 eq.	30	14	86	178
**2**	0.6	150 5 eq.	80	14	86	237

^a^ Reaction conditions: *o*-xylene (0,3 mmol), [Fe(TPFPP)Cl] (1–2 mg), ethanol (2 mL), H_2_O_2_ (5 mol eq.), at RT for 2.5 h; ^b^ H_2_O_2_ added at 2 mol equivalents/h; ^c^ Conversion and selectivity measured by GC-FID analysis; ^d^ Turnover number (TON), two catalytic cycles were considered for product **1b** [35].

The MS spectrum obtained by GC-MS(EI) of the reaction mixtures are reported in Appendix A. Compound **1a** shows [M^+●^] *m*/*z* 122 and loss of CO and CH_3_ fragments as main peaks (Appendix A), matching 3,4-dimethylphenol [36]. The MS spectrum of **1b** shows a di-oxygenated product with [M^+●^] *m*/*z* 138 (Appendix A). Compound **1b** was isolated by fractionation of the reaction mixture using preparative thin layer chromatography (TLC) on silica gel and was fully characterized by ^1^H, ^13^C and 2D-NMR techniques.

The ^1^H NMR spectrum shows the two methyl groups at 1.46 and 2.09 ppm (Figure 2). The latter peak is a doublet with ^4^*J* = 1.5 Hz, due to a four-bonds coupling with H-2. The three signals in the alkene region corroborate the 4-hydroxycyclodienone structure. The H-2 signal is a quintet due long-range coupling with H-6 and CH_3_(C-3). A double doublet is ascribed to H-6 with ^3^*J* = 9.9 Hz and ^4^*J* = 2.0 Hz from coupling with adjacent H-5 and at four-bonds with H-2, respectively. These assignments are corroborated by the HMBC (^1^H^APT) spectrum (Figure 3) and by APT and HSQC spectra in Appendix A.

To our knowledge, this compound has not been previously characterized or isolated and is an attractive building block for synthesis, e.g., as a dienophile in cycloaddition reactions or as a structural analogue of ring C of tetracycline family antibiotics [11]. The different chemoselectivity relative to the previously reported Mn porphyrin catalytic systems, which promote selective alkyl group oxidation [14], highlights this reaction as a new and completely green pathway for the selective functionalization of the aromatic ring of alkylbenzenes.

### 2.2. Quinoline (**2**)

The direct one-pot oxidation of quinoline afforded the quinolone products 3-ethoxy-4-quinolone (**2a**), 4-quinolone (**2b**) and 3-hydroxy-4-quinolone (**2c**). The latter two products were observed also as hydroxyquinoline tautomeric compounds: 4-hydroxyquinoline (**2b***) and 3,4-dihydroxyquinoline (**2c***). The results are collected in Table 2, where for simplicity, selectivity and yield for products **2b** and **2c** corresponds to joint values observed for tautomer compounds (**2b** and **2b***) or (**2c** and **2c***).

Quinoline consumption during the reaction was monitored by GC-FID, but reaction products were not observable by this technique. The reaction mixtures were fractionated by preparative TLC and all the collected fractions were analyzed by NMR and HR-ESI-MS^2^ to obtain products identification/characterization. Subsequently, product selectivity and substrate conversion were obtained by ^1^H NMR analysis of the final reaction mixtures in DMSO-*d*_6_. Similar values of substrate conversion were observed by both techniques for identical reaction conditions (Table 2, entries 1 and 2). Using catalyst loadings of 0.6 and 1.9 mol%, the substrate conversion was 57% and 70%, respectively, after 4 h of reaction time and upon addition of 8 mol equivalents of H_2_O_2_ (Table 2, entries 1 and 3).

The presence of a substituent at 3-position (ethoxy or hydroxy) was dependent on the work-up conditions, namely the temperature of solvent evaporation. Solvent evaporation at 60 °C in the rotary evaporator in Path I and at RT in Path II.

Upon Path I, the fractionation of the reaction mixture by preparative TLC, afforded the 3-ethoxyquinolin-4-(1*H*)-one (3-ethoxy-4-quinolone, **2a**), which was isolated as the single reaction product (Table 2, entry 1).

The ^1^H NMR spectrum of **2a** (Figure 4) shows the selective functionalization on the pyridyl ring, as the four signals in the aromatic region show a multiplicity and coupling pattern typical of a non-functionalized aromatic ring (COSY spectrum in Appendix A). The two doublets at δ 4.01 ppm and 4.73 ppm (broad), coupling with each other (*J* = 4.8 Hz), are ascribed to H-2 and H-1(*N*H), respectively. The low δ values observed for H-2 and C-2 (HSQC spectrum in Figure 5) are expected for 3-substituted-4-quinolones [37], carrying an electron donor substituent at position 3. High electron density on C-2/H-2 is justified by the presence of mesomerism in compound **2a** (Figure 4, upper insert), with a significant contribution of two zwitterionic resonance hybrids to describe its structure. This is confirmed by the multiplet signal at δ 3.91–4.00 ppm, ascribed to ethoxy -CH_2_ group. The contribution of the two zwitterionic structures leads to a hindrance in Ar-OEt bond rotation, resulting in distinct chemical environment on the -CH_2_ protons [38,39]. The hydroxyl tautomer of compound **2a**, 3-ethoxy-4-hydroxyquinoline, was not observed.

We found no previous references in the literature for compound **2a**, and this methodology may be an effective and green way to produce new 3-substituted quinolone derivatives, by direct functionalization of quinoline.

Upon Path II (solvent evaporation at RT, 17–22 °C), the fractionation of the reaction mixture by preparative TLC, afforded quinolin-4(1*H*)-one (4-quinolone, **2b**) and 3-hydroxyquinolin-4(1*H*)-one (3-hydroxy-4-quinolone, **2c**) and the corresponding tautomers 4-hydroxyquinoline (**2b***) and 3,4-dihydroxyquinoline (**2c***). The selectivity was 46% and 22% for the mixtures of tautomers (**2b** + **2b***) and (**2c** + **2c***), respectively (Table 2, entry 2).

Compounds **2b** and **2b***, **2c** and **2c*** were identified by HR-ESI-MS^2^ (Appendix A). The [M + H]^+^ ions in the MS spectra were *m*/*z* 146.060 and *m*/*z* 162.055 for compounds **2b**/**2b*** and **2c**/**2c***, respectively. NMR studies in DMSO-d6 (^1^H, APT, COSY, and HSQC; Appendix A) confirmed the identification of these compounds.

The ^1^H NMR spectrum of the total reaction, after evaporation at RT (Path II, Table 2, entry 2), was obtained in DMSO-d6 (Appendix A). The area of chosen non-overlapping peaks from quinoline and reaction products were used for quantification of product selectivity, product yield and substrate conversion. The 4-hydroxy-tautomers were observed as the major products. It should be noted that the ferric center of [Fe(TPFPP)Cl] has a markedly acidic character [40] and may confers acidity to the reaction media, favoring the presence of hydroxyl-tautomers **2b*** and **2c***.

### 2.3. Acridine (**3**)

Acridine oxidation yielded 1:2-epoxyacridine (**3a**) and *anti*-1:2,3:4-diepoxyacridine (**3b**). NMR studies of the reaction mixture after chromatographic separation of the catalyst allowed compounds’ identification by comparison with previously described data [12]. Better selectivity was obtained for compound **3a** (90%, Table 1, entry 1) than in those studies using the MnP catalytic system (70%, Figure 1) [12].

As acridine has an N atom in the structure, which confers basicity to the substrate that might influence the oxidation reaction, one reaction was carried out with addition of HNO_3_. The results are presented in Table 3. It is observed that the pH does not lead to significant changes in the conversion of acridine, resulting only in a small increase in the yield of the monoepoxide (**3a**).

According to the ^1^H spectrum of the reaction mixture, shown in Figure 6, there is a total of 18 protons, which indicates the presence of a mixture of the two products, **3a** and **3b**. The only two singlets present in the spectrum, at δ 8.38 and 8.20 ppm, correspond to H-9 of both compounds and their areas were used to quantify products selectivity. The ^1^H, APT, and COSY NMR spectra are reported in Appendix A.

### 2.4. Catalyst Stability

The reactions were followed by UV–vis. At the beginning of the reaction, the Fe porphyrin Soret band, at 410 nm, is observed and its intensity decreases as the reaction proceeds. This indicates the concomitant oxidation of the porphyrin macrocycle (Appendix A). The cessation of substrate conversion relates to the complete disappearance of the Soret band after the TON maximum of 237, 61, and 285 for *o*-xylene, quinoline, and acridine, respectively.

### 2.5. Considerations on the Mechanism

Catalytic performance of the [Fe(TPFPP)Cl] (FePF) might be associated with the typical acidic character of iron porphyrins [40], which is intensified by the strong electron-withdrawing porphyrin ligand due to extensive fluorination. The iron-hydroperoxy-species [PFe(III)-OOH] formed by coordination of hydrogen peroxide to the metal center and subsequent deprotonation, has been considered the active oxidant in the catalytic cycle. In the absence of a co-catalyst, it is not expected that this species evolve into an oxo species [32].

Metallo-hydroperoxy species have been described as the active oxidant in epoxidation reactions or in the generation hydroxyl radicals [3]. Previous studies showed that the Fe(III) porphyrin is effective in alkene epoxidation and aromatic hydroxylation and an EPR spin trap study confirmed the absence of free hydroxyl radicals in these conditions [3]. The [PFe-OOH] is a strong oxidant, which can be further activated in the presence of a protic solvent by hydrogen bond formation with the hydroperoxide group [41]. This leads to an enhanced δ^+^ at the distal oxygen (Figure 3, central species).

In the present work, it was observed that, unlike Mn porphyrins in aprotic solvent (acetonitrile) and with a co-catalyst, the present [FePF@H_2_O_2__EtOH] system performs the selective aromatic oxidation of *o*-xylene, the oxidation of the aza-ring of quinoline and the epoxidation of the peripheral ring of acridine.

In previous studies [31], it was pointed out that the action of this catalyst in the formation of naphthoquinone by naphthalene oxidation (Figure 1D) can be explained by the formation of naphtol via an electrophilic substitution mechanism (Figure 3A, path ii). The electrophilic attack of [Fe-OO(δ^+^)H] on the aromatic ring π-system, with formation of a carbocation intermediate and recovery of aromaticity by deprotonation of the adjacent proton.

However, the results of the present work suggest a direct epoxidation of the aromatic π-system (Figure 3A, path i) by an addition reaction, and the hydroxylated derivatives formed result from subsequent rearrangement of the epoxide ring in acidic medium (path iii).

The presence of path (i) is confirmed by: (a) isolation of acridine epoxides; (b) formation of the quinoline derivatives **2a** (3-ethoxy-4-quinolone) and **2c** (3-hydroxy-4-quinolone) (Figure 3C), either resulting by epoxide ring opening through nucleophilic attack of EtOH (60 °C) or H_2_O (RT); (c) regioselectivity of hydroxylation in the formation of the final xylene product **1b** (Figure 3B), suggesting epoxidation of the aromatic ring and not electrophilic substitution (path ii), as in the latter case, the derivatives should result from the formation of a tertiary carbocation intermediate, namely, 1,3-dihydroxy-4,5-dimethylbenzene or 3,4-dimethyl-*o*-benzoquinone.

The oxidation of intermediate **I** might occur also non-catalytically in the oxidizing reaction media, similarly to the formation of benzoquinones from hydroquinones previously observed [14].

## 3. Materials and Methods

### 3.1. Materials

The chloro [5,10,15,20-tetraquis(pentafluorophenyl)porphyrinate] iron (III) [Fe(TPFPP)Cl] (FePF) was prepared by a literature procedure, in environmentally compatible conditions, using microwave heating [31]. Quinoline (98%), acridine (97%), and H_2_O_2_ 30% *w*/*w* were purchased from Sigma-Aldrich (St. Louis, MO, USA). *o*-Xylene (98%) *n*-hexane, and ethyl acetate were acquired from Fisher Scientific (Waltham, MA, USA). Ethanol was from AppliChem (Gatersleben, Germany). All solvents were p.a. grade. Nitric acid (65%) was from PanReac (Barcelona, Spain). The chromatographic purifications were carried out using silica gel 60 F254 from Merck (Darmstadt, Germany).

### 3.2. Instrumentation

The GC-FID analyses were performed using a Varian 3900 chromatograph (Palo Alto, CA, USA), using nitrogen as carrier, and GC-MS analyses were performed in a Thermo Scientific Trace 1300, coupled to a Thermo Scientific ISQ Single quadropole MS apparatus (Waltham, MA, USA), using helium as the carrier gas. In both cases, DB-5-type-fused silica Supelco (Sigma-Aldrich) capillary columns were used (30 m, 0.25 mm i.d.; 0.25 μm film thickness) and the temperature program was: 70 °C (1 min), 20 °C min^−1^ to 200 °C (5 min). The injector temperature was set at 200 °C and the detector temperature was set at 250 °C.

UV–vis absorption spectra were recorded at room temperature on a Genesys 10s Thermo Scientific Spectrophotometer in the region 300–800 nm.

High-resolution electrospray ionization mass spectra (HR-ESI-MS) were obtained using an LTQOrbitrap XL mass spectrometer (Thermo Scientific). Evaporated samples were dissolved in acetonitrile while reaction mixtures were directly injected and infused into the electrospray ion source at 10 μL·min^−1^. The spectrometer was operated in the positive ionization mode with the capillary voltage set to +3.1 kV, sheath gas flow to 6, and the temperature of the ion transfer capillary to 275 °C.

NMR spectra (1D and 2D) were recorded on Bruker Avance instruments operating at a frequency of 400 MHz for ^1^H experiments and 100 MHz for ^13^C experiments, with sample temperatures of 22 °C and using CDCl_3_ or DMSO-*d*_6_ as solvent (Euroisotop, Cambridge, UK).

NMR and MS analyses were performed at CEMUP (Centro de Materiais da Universidade do Porto).

### 3.3. Catalytic Oxidation Reactions

The catalytic experiments were performed using the following general procedure: The substrate (0.3 mmol), the catalyst [Fe(TPFPP)Cl] (FePF), from 0.3 mol % (1 mg, 1 μmol) to 1.9 mol% (5 mg, 5 μmol), as indicated in Table 1, Table 2 and Table 3, were dissolved in 2 mL of ethanol and stirred at RT (17–22 °C) protected from light. H_2_O_2_ (aq.) 30% *w*/*w* was progressively added to the reaction mixture, by addition of aliquots of 0.5 mol equivalents relatively to the substrate every 15 min. The reactions were terminated when the substrate conversion did not change despite the addition of H_2_O_2_. When specified, the acridine reaction media was acidified by addition of HNO_3_ until pH~4.6.

At the end of reactions, the solvent was evaporated at 60 °C in the rotary evaporator (Work-up-1, used for quinoline) or at RT (Work-Up 2, used for all substrates) and the reaction mixtures were separated by TLC, using mixtures of ethyl acetate: *n*-hexane as eluent: (40:60% *v*/*v*) for xylene (**1**) and (50:50% *v*/*v*) for quinoline (**2**) and acridine (**3**). Compounds were revealed on TLC plates using a UV lamp and were removed from silica with the same eluent used for chromatography.

For ^1^H NMR analyses of the total reaction mixtures, the final reaction was passed through a small plug of silica-gel and eluted with DMSO-*d*_6_.

Conversion (%) = [n (sum of products)/n of substrate]; Selectivity (%) = [n of product P/n (sum of products)]; Yield (%) = [n of product P/n of substrate].

### 3.4. Spectroscopic Data of Products

4-hydroxy-3,4-dimethylcyclohexa-2,5-dienone (**1b**) ^1^H NMR (CDCl_3_, 400 MHz) δ 1.46 (3H, s, CH_3__C-4), 2.09 (3H, d, *J* = 1.5 Hz, CH_3__C-3), 6.01 (1H, quintet, *J* = 1.5 Hz, H-2), 6.12 (1H, dd, *J* = 9.9, 2.0 Hz, H-6), 6.88 (1H, d, *J* = 9.9 Hz, H-5); ^13^C NMR (CDCl_3_, 400 MHz) δ 18.0 (CH_3__C-3), 26.0 (CH_3__C-4), 69.2 (C, C-4), 125.9 (CH, C-2), 127.0 (CH, C-6), 152.5 (CH, C-5), 161.7 (C, C-3), 185.8 (C, C-1); EIMS *m*/*z* (relative abundance %) 138 [M]^+•^ (43), 123 (100), 110 (54), 95 (77).

3-ethoxy-4-quinolone (**2a**) ^1^H NMR (CDCl_3_, 400 MHz) δ 1.00 (3H, t, *J* = 7.1 Hz, -CH_3_), 3.96 (2H, m, -CH_2_), 4.01 (1H, d, *J* = 4.8 Hz, H-2), 4.73 (1H, d-broad, *J* = 4.8 Hz, H-1), 7.52 (1H, t, *J* = 8.2 Hz, H-6), 7.70 (1H, t, *J* = 8.0 Hz, H-7), 7.85 (1H, d, *J* = 8.0 Hz, H-8), 8.18 (1H, dd, *J* = 8.2, 1.3 Hz, H-5); ^13^C NMR (CDCl_3_, 400 MHz) δ 14.1 (-CH_3_), 54.6 (CH, C-2), 60.8 (-CH_2_), 124.6 (CH, C-5), 129.3 (CH, C-6), 130.1 (CH, C-8), 133.6 (CH, C-7).

4-quinolone (**2b**) ^1^H NMR (DMSO-*d*_6_, 400 MHz) δ 6.51 (1H, d, *J* = 9.0 Hz, H-3), 7.19 (1H, t, *J* = 7.8 Hz, H-6), 7.32 (1H, d, *J* = 7.9 Hz, H-8), 7.50 (1H, t, H-7), 7.66 (1H, d, *J* = 7.8 Hz, H-5), 7.91 (1H, d, *J* = 9.0 Hz, H-2); ^13^C NMR (DMSO-*d*_6_, 400 MHz) δ 115.6 (CH, C-8), 122.2 (CH, C-3), 122.3 (CH, C-6), 128.4 (CH, C-5), 130.9 (CH, C-7), 140.8 (CH, C-2), HRESIMS *m*/*z* (relative abundance %) 162.055 [M + H]^+^, (100), 144.045 (7), 134.060 (7), 116.050 (10).

4-hydroxyquinoline (**2b***) ^1^H NMR (DMSO-*d*_6_, 400 MHz) δ 7.50 (1H, d, *J* = 7.3 Hz, H-3), 7.76 (1H, t, *J* = 7.5 Hz, H-6), 7.84 (1H, t, *J* = 8.4 Hz, H-7), 7.99 (1H, d, *J* = 7.3 Hz, H-2), 8.11 (1H, d, *J* = 7.5 Hz, H-5), 8.55 (1H, d, *J* = 8.4 Hz, H-8); ^13^C NMR (DMSO-*d*_6_, 400 MHz) δ 119.3 (CH, C-8), 122.4 (CH, C-3), 126.1 (CH, C-2), 129.1 (CH, C-5), 129.3 (CH, C-6), 130.9 (CH, C-7), 119.6 (C, C-4a), 141.2 (C, C-8a), 162.6 (C, C-4); HRESIMS *m*/*z* (relative abundance %) 146.060 [M + H]^+^, (100), 129.057 (2), 128.050 (3).

## 4. Conclusions

Inert aromatic compounds have been selectively functionalized by catalytic oxidation in mild and green conditions, using H_2_O_2_ as green oxidant, ethanol as green solvent, in the absence of other additives, at room temperature and using an electron withdrawing iron porphyrin catalyst, obtainable in eco-sustainable conditions, and used in a low loading of <2 mol%. The results support the occurrence of an initial direct epoxidation of the aromatic ring, leading to an unusual selectivity for *o*-xylene oxidation, as it occurred exclusively on the aromatic ring and not on the methyl groups, as previously observed. Moreover, the functionalization of acridine on the peripheral ring, instead of on the *meso* 9-position was unusual. The new methodology can be very attractive for the preparation of new 3-substituted quinolone derivatives, which have high potential for biological activity. The oxidations resulted in loss of aromaticity in products or in one of the aromatic rings. Two new compounds with attractive application potential were isolated and characterized.

The results point to the future relevance of aromatic epoxidation reactions, still largely unexplored in organic synthesis. This is mainly of importance in the valorization of aromatic products resulting from recycling processes based on pyrolysis of biomass and waste. Further developments of the catalytic system can be pursued developing more easily obtainable highly electron withdrawing iron catalysts.

## Data Availability

Not applicable.

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
