# Peer review of "Green Aromatic Epoxidation with an Iron Porphyrin Catalyst for One-Pot Functionalization of Renewable Xylene, Quinoline, and Acridine"

_molecules, 2023, doi:10.3390/molecules28093940_

Round 1

Reviewer 1 Report

Comments/Questions:

Page 1 Ln 27: cytochrome P450.

Page 2 Ln 42: environment.

Page 2 Ln 53: Alzheimer’s.

Page 2 Ln 53: Green acridine?

Page 2 Ln 60-63: rewrite the sentence.

Page 2 Ln 67: in presence of.

Are the Fe-porphyrin catalyst in Scheme 2 and Scheme 1D same? If so, use FePF in Scheme 1D.

Page 3: Scheme 2 Include the compound # for the reactants and products consistent with the text. Did the authors subjected 1a to the catalytic oxidation conditions to see if in fact 1a is converted to 1b?

Page 4 Ln 120: 2D-NMR instead of bi-dimensional.

It would be helpful to include 1b structure from Table 1 as a part of Figure 2. This will be helpful for the reader.

Page 5 Ln 126: There is no intramolecular hydrogen bond in the system. It is just J coupling.

What are the yields of 4-quinolone products? (as efficiency is understood by the product yield)

Page 7 Ln 178: There is no hydrogen bond between CH2 and C=O. The compound can aromatize to 3-ethoxy-4-hydroxyquinoline by tautomerization.

Figure 5: It is hard to see the hydrogen labels and carbon labels in HSQC.

Page 7 Ln 174-185: Difficult to understand what authors are trying to explain.

Scheme 3: 2b and 2b* structures are incorrect. Tautomerization is not represented by resonance arrow etc.

Scheme 3C: the charges on the N are wrong trough out.

Pages 14-16: References: journal abbreviation needs to be fixed (almost all the reference). For instance Chem. Rev. instead of chemical reviews.

Overall, not well written and major revision is required and cannot be recommended for publication in the current form.

Reviewer 2 Report

1.       An abstract must be fully self-contained and make sense by itself, without further reference to outside sources or to the actual paper. It is important to provide the relevance or importance of your work and the main outcomes. Apart from this, the abstract is up to the mark and concisely written with a complete short introduction of the manuscript.

2.       The literature from past work done in the same field missing to strengthen the introduction section.

3.       The strong hypothesis, scientific facts, and validation of previous reports are entirely missing. You must rewrite it and cite the recommended papers listed in the comment section of this review report.

4.       Compare your material's efficiency with other published articles for the same target.

5.       The conclusion section failed to enlighten the spirit of the finding and is missing the results. Rewrite it.

Reviewer 3 Report

In the present study the authors describe the application of the [FePF@H2O2_EtOH] catalytic system, at room temperature (RT), in the oxidative valorization of the renewable aromatic compounds, but there are some points that needs to be addressed before consideration of this publication in this journal.

1.     Why the authors called epoxidation of aromatic compounds as green one as indicated in the title and in the abstract? The reviewer wants to know how this project is related to the green one. Are the authors wants to say that by products obtained from the epioxidation donot cause pollution or they can be used again and again?

2.     The author mentioned the figure number SM15,SM16 in the section of catalyst stability but these figures are not found in the manuscript and supporting information. The figures numbers should be revised in the whole manuscript carefully, and wisely.

3.     In the section catalytic oxidation reactions, the authors mentioned the TLC ratio as  (40:60) for xylene (1) and (50:50) for quinoline (2) and acridine (3). The reviewer wants to know the ratio in ml please justify that the authors take this amount in ml or not ?

4.     Most of proton NMR figures need to be revised again as they are not clear to see as given in the figure 4, and 5.The whole of manuscript therefore needs to be revised accordingly.

5.     The structure of fluorinated iron porphyrin abbreviated as [Fe(TPFPP)Cl] (FePF) is given in the  scheme 2 needs to be revised again to justify the position of fluorinated aromatic ring.

6.     The FT-IR spectra and UV-Visible spectra of all the synthesized compounds and separated compounds should be provided.

7.     Most of the NMRs peaks given in the spectra should be mentioned as in the figure 4 the peak at 1.5 ppm. The authors should clearly mentioned the peak with strong intensity. keeping in view of the above point the whole of the manuscript needs to  be  revised accordingly.

Round 2

Reviewer 1 Report

The authors have answered this reviewer’s comments and edited the document and presented the data appropriately. The article can be recommended for publication.

Reviewer 3 Report

 In the current version of the manuscript entitled Green Aromatic Epoxidation with an Iron Porphyrin Catalyst for One Pot Functionalization of Renewable Xylene, Quinoline and Acridine, Given the proper revisions made by authors, I comment the publication of the work. Meanwhile, I also have no objection to other reviewer(s) or editor making other decisions on this issue.